# Point-of-care prognostication in moderate Covid-19: Analytical validation and prognostic accuracy of a soluble urokinase plasminogen activator receptor (suPAR) rapid test

**Arjun Chandna**[1,2]*, **Raman Mahajan**[3], **Priyanka Gautam**[4], **Lazaro Mwandigha**[5], **Sabine Dittrich**[2,6,7], **Vikash Kumar**[3], **Jennifer Osborn**[6], **Pragya Kumar**[8], **Constantinos Koshiaris**[5], **George M. Varghese**[4], **Yoel Lubell**[2,9], **Sakib Burza**[3,10]

1 Cambodia Oxford Medical Research Unit, Angkor Hospital for Children, Siem Reap, Cambodia, 2 Centre for Tropical Medicine & Global Health, University of Oxford, Oxford, United Kingdom, 3 Médecins Sans Frontières, New Delhi, India, 4 Department of Infectious Diseases, Christian Medical College, Vellore, India, 5 Nuffield Department of Primary Care Health Sciences, University of Oxford, Oxford, United Kingdom, 6 Foundation for Innovative Diagnostics, Geneva, Switzerland, 7 Deggendorf Institut of Technology, European-Campus Rottal Inn, Pfarrkirchen, Germany, 8 Department of Community & Family Medicine, All India Institute of Medical Sciences, Patna, India, 9 Mahidol-Oxford Tropical Medicine Research Unit, Mahidol University, Bangkok, Thailand, 10 Department of Clinical Research, London School of Hygiene & Tropical Medicine, London, United Kingdom

* arjun@tropmedres.ac, arjunchandna@gmail.com

**Data Availability Statement:** De-identified, individual participant data from this study will be

## Abstract

The soluble urokinase plasminogen activator receptor (suPAR) has been proposed as a biomarker for risk stratification of patients presenting with acute infections. However, most studies evaluating suPAR have used platform-based assays, the accuracy of which may differ from point-of-care tests capable of informing timely triage in settings without established laboratory capacity. Using samples and data collected during a prospective cohort study of 425 patients presenting with moderate Covid-19 to two hospitals in India, we evaluated the analytical performance and prognostic accuracy of a commercially-available rapid diagnostic test (RDT) for suPAR, using an enzyme-linked immunosorbent assay (ELISA) as the reference standard. Our hypothesis was that the suPAR RDT might be useful for triage of patients presenting with moderate Covid-19 irrespective of its analytical performance when compared with the reference test. Although agreement between the two tests was limited (bias = -2.46 ng/mL [95% CI = -2.65 to -2.27 ng/mL]), prognostic accuracy to predict supplemental oxygen requirement was comparable, whether suPAR was used alone (area under the receiver operating characteristic curve [AUC] of RDT = 0.73 [95% CI = 0.68 to 0.79] vs. AUC of ELISA = 0.70 [95% CI = 0.63 to 0.76]; p = 0.12) or as part of a published multivariable prediction model (AUC of RDT-based model = 0.74 [95% CI = 0.66 to 0.83] vs. AUC of ELISA-based model = 0.72 [95% CI = 0.64 to 0.81]; p = 0.78). Lack of agreement between the RDT and ELISA in our cohort warrants further investigation and highlights the importance of assessing candidate point-of-care tests to ensure management algorithms reflect the assay that will ultimately be used to inform patient care. Availability of a quantitative

available to researchers whose proposed purpose of use is approved by the data access committees at Médecins Sans Frontières and the Mahidol-Oxford Tropical Medicine Research Unit. Inquiries or requests for the data may be sent to data. sharing@london.msf.org and datasharing@tropmedres.ac. Researchers interested in accessing biobanked samples should contact the corresponding authors who will coordinate with the respective institutions.

**Funding:** The PRIORITISE (Prognostication of Oxygen Requirement in Patients with Non-severe SARS-CoV-2 Infection) study was funded by Médecins Sans Frontières, India. The Wellcome Trust provides core funding to the Mahidol-Oxford Tropical Medicine Research Unit in Bangkok [220211 to MORU; 215604/Z/19/Z to YL], which supported the design, monitoring and analysis of the study. The suPAR RDTs were procured by FIND with funding from the Australian Government. CK is supported by a Wellcome Trust/Royal Society Sir Henry Dale Fellowship [211182/Z/18/Z]. Médecins Sans Frontières maintained a sponsor/investigator role for the study. All other funders had no role in study design, data collection and analysis, decision to publish, or preparation of the manuscript. For the purpose of open access, the author has applied a CC BY public copyright license to any Author Accepted Manuscript version arising from this submission.

**Competing interests:** I have read the journal's policy and the authors of this manuscript have the following competing interests: Sabine Dittrich and Jennifer Osborn declare that they are employed by FIND. All other authors have declared that no competing interests exist.

point-of-care test for suPAR opens the door to suPAR-guided risk stratification of patients with Covid-19 and other acute infections in settings with limited laboratory capacity.

## Introduction

In busy clinical settings the window for effective triage is short. Biochemical biomarkers included in risk stratification tools require rapid turnaround times. However, studies developing triage tools commonly use laboratory-based platforms to quantify biomarker concentrations, the accuracy of which may differ from point-of-care tests required to inform timely management of individual patients in settings with limited laboratory capacity [1].

The soluble version of the urokinase plasminogen activator receptor (suPAR) is upregulated during the host response to infection [2]. Measurements of suPAR have been shown to be useful in the diagnosis and prognosis of a wide range of infections and infectious syndromes [3–8], including SARS-CoV-2 [9,10]. Recently, our group and others developed clinical prediction models incorporating suPAR for both community- and hospital-based triage of patients with Covid-19 [11,12]. However, although suPAR is measurable using a commercially-available rapid test, these studies quantified suPAR using laboratory-based immunoassays.

We performed an analytical validation of a rapid diagnostic test (RDT) for suPAR using samples from a multi-centre prospective cohort study conducted in India [11]. We evaluated the prognostic accuracy of the RDT by comparing the predictive performance of suPAR quantified using the RDT and an enzyme-linked immunosorbent assay (ELISA). Our hypothesis was that the suPAR RDT might be useful for triage of patients presenting with moderate Covid-19 irrespective of its analytical performance when compared with the reference test.

## Methods

### Ethics statement

Ethical approval was given by the All India Institute for Medical Sciences Patna Ethics Committee; Christian Medical College Ethics Committee; Oxford Tropical Research Ethics Committee; and Médecins Sans Frontières Ethical Review Board. All participants provided informed written consent prior to recruitment.

### Study population and clinical data collection

This is a secondary analysis of data collected during a prospective cohort study conducted at two hospitals in India between October 2020 and July 2021. The study design and setting have been described previously [11]. Briefly, consenting consecutive adults (aged $\geq$ 18 years) presenting with clinically-suspected SARS-CoV-2 infection of moderate severity (defined as peripheral oxygen saturation ($SpO_2$) $\geq$ 94% and respiratory rate (RR) $\leq$ 30 breaths per minute (BPM) in the context of systemic symptoms) were recruited. Clinical parameters were measured at enrolment. Admitted participants were followed-up each day until death, discharge or day 14, whichever occurred first. For those not admitted or discharged prior to day 14, follow-up was conducted via telephone on days 7 and 14, with participants who reported worsening and/or persistent symptoms recalled to have their $SpO_2$ and RR measured. The primary outcome for the original study was development of a supplemental oxygen requirement within 14 days of enrolment, defined as any of: $SpO_2 < 94\%$; $RR > 30$ BPM; $SpO_2/FiO_2 < 400$; or death.

## Laboratory procedures

Venous blood samples were collected in ethylenediaminetetraacetic acid (EDTA) tubes at enrolment, centrifuged within four hours, and plasma aliquots stored at ≤ -20˚C. Frozen plasma aliquots were transported on dry ice and thawed at 2-8˚C overnight prior to analysis. suPAR concentrations were quantified using the suPARnostic ELISA and the suPARnostic Quick Triage test (Virogates, Denmark). The suPARnostic ELISA is a simplified double monoclonal antibody sandwich assay which requires 15μL of plasma. Samples were run in duplicate and mean concentration reported. The Quick Triage test is a RDT based on lateral flow principles. It requires 10μL of plasma and has a dynamic range of 2–15 ng/mL. Paired with an automated lateral flow optical reader it has a time-to-result of 20 minutes. Both tests were performed as per the manufacturer's instructions using the same aliquot of thawed plasma [13,14], and the operators who performed both tests were blinded to the results of the other test.

## Primary and secondary outcomes

The primary outcome was the prognostic accuracy of the RDT (index test) assessed by comparing the predictive performance (area under the receiver operating characteristic curve; AUC) of the RDT to the ELISA (reference standard). The secondary outcome was the analytical performance of the RDT, assessed by quantifying the agreement between the RDT and ELISA.

## Statistical methods

Logistic regression was used to quantify the AUC and compare (DeLong method) the prognostic accuracy of the RDT and ELISA (R package: *pROC*) [15,16]. The analytical performance of the RDT was evaluated using a Bland-Altman plot to estimate the bias and limits of agreement between the RDT and the ELISA (R package: *blandr*) [17,18]. Assessment of agreement was limited to samples within the dynamic range (2–15 ng/mL) of the RDT. A sensitivity analysis was conducted where samples quantified on the ELISA but outside the dynamic range of the RDT were set to the limits of quantification of the RDT. All analyses were performed in R, versions 4.0.2 and 4.0.3 [19].

## Sample size

For the purposes of an analytical validation, the Clinical and Laboratory Standards Institute (CLSI) recommend a minimum sample size of 100 to evaluate agreement between a candidate and reference test [20]. To maximise precision of the results, we used all available samples from the original study (n = 425).

## Study reporting

This investigator-initiated study was prospectively registered (ClinicalTrials.gov; NCT04441372), with protocol and statistical analysis plan uploaded to the Open Science Framework platform (DOI: 10.17605/OSF.IO/DXQ43). The study is reported in accordance with the Standards for Reporting Diagnostic accuracy studies (STARD) guidelines (Table A in S1 Text) [21].

# Results

## Clinical outcomes

Between 22 October 2020 and 3 July 2021, 426 participants were recruited of whom 425 had suPAR concentrations quantified using the RDT and ELISA. Three participants were lost-to-

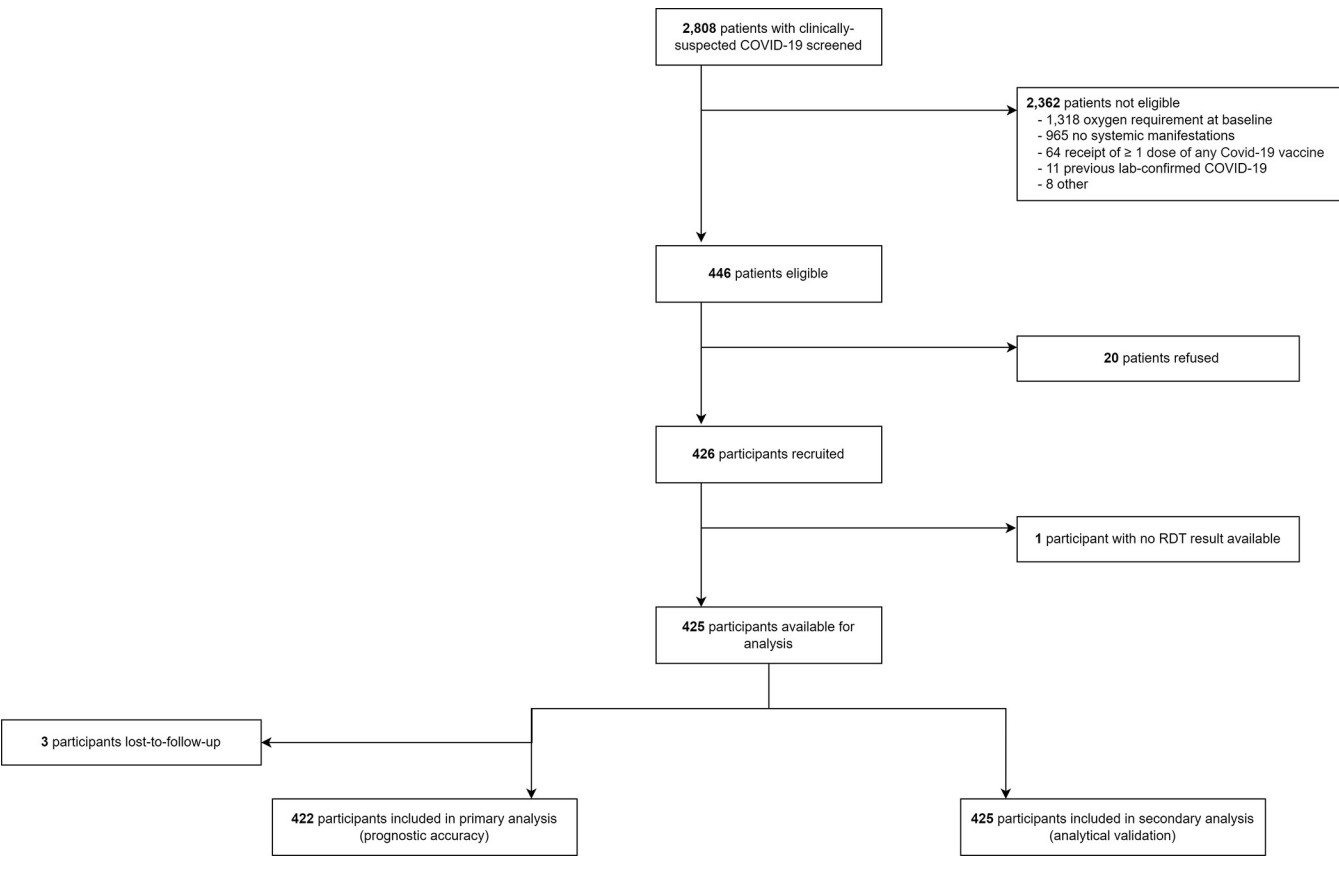

**Fig 1. Eligibility of samples for inclusion in analytical validation and prognostic accuracy evaluation.**

follow-up before day 14, leaving 422 participants available for the primary analysis (Fig 1). Eighty-nine participants developed a supplemental oxygen requirement (89/422; 21.1%).

## Limited agreement between the reference ELISA and candidate RDT for quantification of suPAR

Forty-four samples returned values outside the dynamic range of the RDT on either the ELISA and/or RDT, leaving 381 paired samples for assessment of agreement. Median suPAR concentration was higher when quantified by the RDT (Table 1; 6.6 vs. 4.2 ng/mL; $p < 0.001$). The two tests were correlated (Pearson's correlation = 0.66 [95% CI = 0.60–0.71]; $p < 0.001$) but there was limited agreement, with the RDT returning higher values than the ELISA on average. A Bland-Altman plot indicated a bias of -2.46 ng/mL (95% CI = -2.65 to -2.27 ng/mL) with upper and lower limits of agreement of 1.21 ng/mL (95% CI = 0.89 to 1.54 ng/mL) and -6.13 ng/mL (95% CI = -6.45 to -5.81 ng/mL) respectively (Fig 2). A sensitivity analysis in which samples outside the dynamic range of the RDT were set to the limits of detection of the RDT returned similar results (Fig A in S1 Text). Given the disagreement with the reference test we verified the reproducibility of the RDT results by repeating the measurements of all participants at one site (n = 125) using another batch of RDTs (Fig B in S1 Text).

Median values (IQR) are reported for continuous variables. *3 participants missing information about supplemental oxygen requirement excluded from table but included in assessment of agreement. †Wilcoxon rank sum test.

**Table 1. Baseline characteristics of the cohort, stratified by progression to supplemental oxygen requirement.**

| Baseline characteristics | Overall (n = 378)* | Developed oxygen requirement | | |
|---|---|---|---|---|
| | | No (n = 297) | Yes (n = 81) | p-value[†] |
| **Demographic characteristics** | | | | |
| Age (years) | 54.0 (42.0, 63.0) | 54.0 (42.0, 62.0) | 54.0 (42.0, 67.0) | 0.70 |
| Male sex | 261 / 378 (69%) | 199 / 333 (67%) | 62 / 81 (77%) | 0.10 |
| **suPAR assay** | | | | |
| suPAR ELISA (ng/ml) | 4.2 (3.2 to 5.6) | 4.0 (3.1 to 5.3) | 5.2 (3.8 to 6.4) | < 0.001 |
| suPAR RDT (ng/ml) | 6.6 (5.2 to 8.5) | 6.2 (5.0 to 8.3) | 8.0 (6.8 to 9.4) | < 0.001 |

## Prognostic accuracy is maintained when a rapid point-of-care test is used to quantify suPAR concentrations instead of a laboratory-based assay

Although agreement between the two tests was limited, we recognised that the RDT might still have utility providing predictive performance did not deteriorate when suPAR concentrations were quantified using the RDT rather than the ELISA. Participants who progressed to develop a supplemental oxygen requirement had higher median baseline suPAR levels, irrespective of the assay used for quantification (Table 1; RDT = 8.0 vs. 6.2 ng/mL, p < 0.001; ELISA = 5.2 vs. 4.0 ng/mL, p < 0.001). Prognostic accuracy of the RDT was comparable to the ELISA (Fig 3; AUC of RDT = 0.73 [95% CI = 0.68 to 0.79] vs. AUC of ELISA = 0.70 [95% CI = 0.63 to 0.76]; p = 0.12) for discriminating participants who would progress to require supplemental oxygen. Evaluation of the prognostic accuracy of the RDT within the framework of our previously

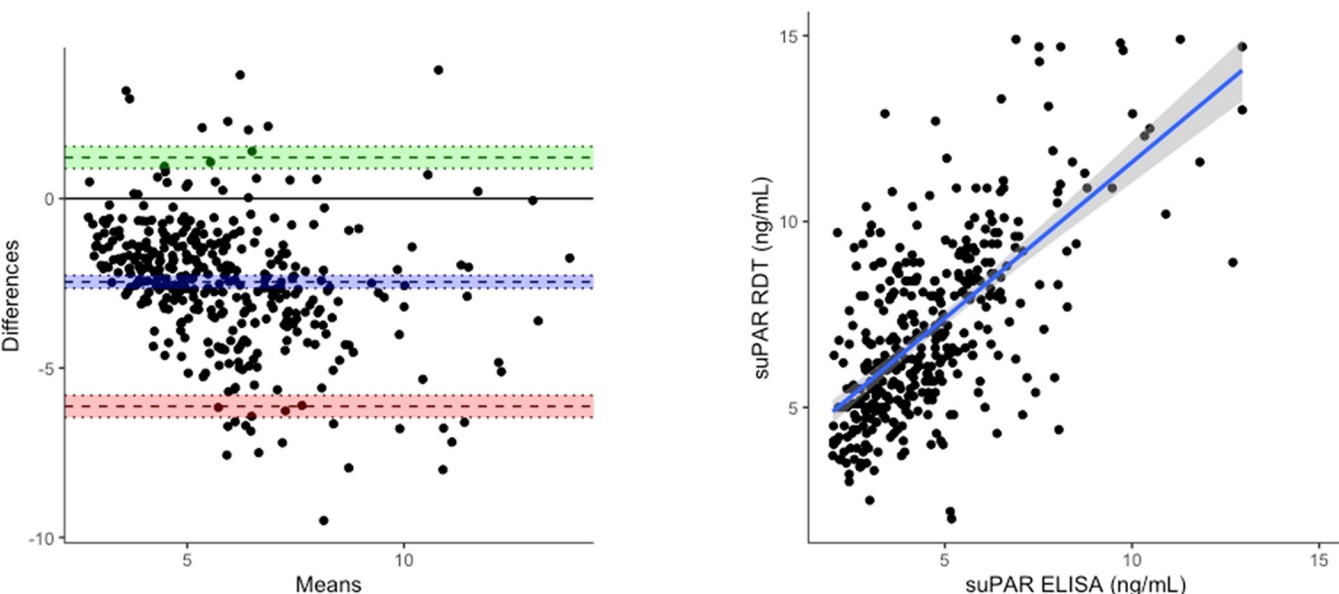

**Fig 2. Relationship between the suPAR RDT and ELISA.** Left panel: Bland-Altman plot indicating agreement between the two tests. Difference between RDT and ELISA measurement in ng/mL plotted on y-axis. Mean of the RDT and ELISA measurement in ng/mL plotted on x-axis. Limits of agreement defined by the concentrations within which 95% of the data fall. Blue line indicates bias, green line indicates upper limit of agreement, red line indicates lower limit of agreement, all with 95% confidence intervals. Right panel: Scatterplot indicating correlation between the two tests.

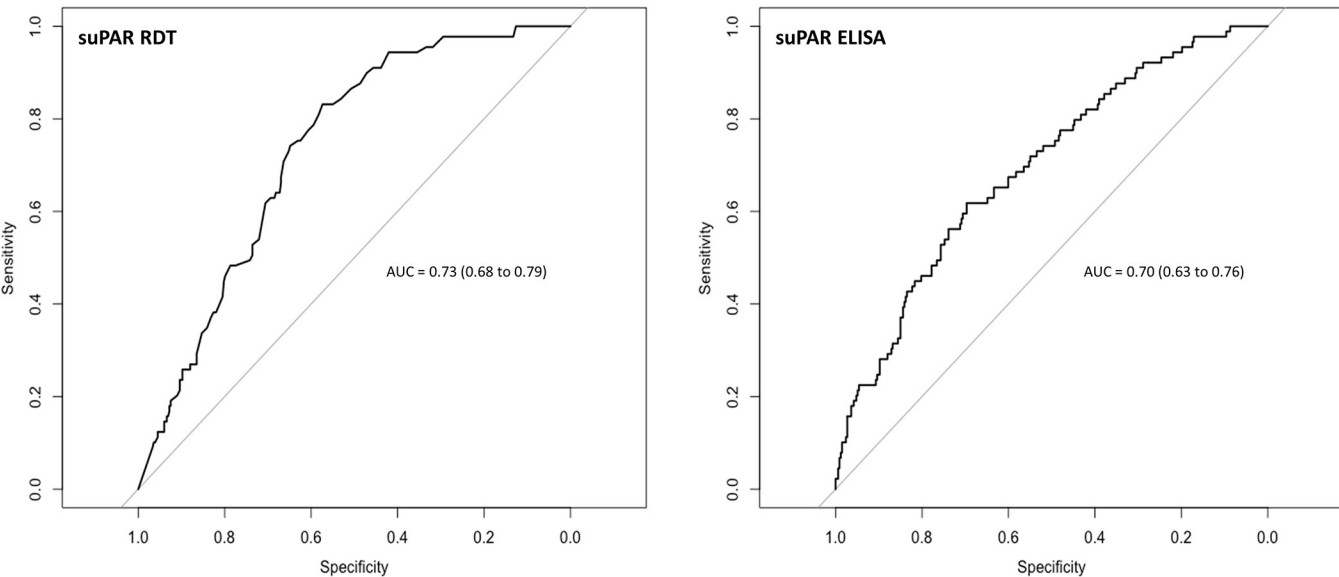

**Fig 3. Prognostic accuracy of suPAR quantified using a point-of-care RDT or laboratory-based ELISA.** AUC = area under the receiver operating characteristic curve.

published multivariable clinical prediction model confirmed comparable predictive performance of the model whether suPAR was quantified using the RDT or the ELISA (AUC of RDT-based model = 0.74 [95% CI = 0.66 to 0.83] vs. AUC of ELISA-based model = 0.72 [95% CI = 0.64 to 0.81]; p = 0.78). The weighting (regression coefficient) for suPAR within the model differed depending on which assay was used for quantification (Text A in S1 Text).

## Discussion

We demonstrate equivalent prognostic accuracy between an index RDT for suPAR and a laboratory-based ELISA to predict progression to supplemental oxygen requirement amongst patients presenting with moderate Covid-19. Comparable predictive performance of the RDT was achieved despite limited agreement with the reference test and was maintained whether suPAR was used alone or as a constituent parameter in a previously published multivariable clinical prediction model for the triage of patients with moderate Covid-19 [11].

Our results illustrate the importance of conducting both analytical and clinical validations of candidate point-of-care tests. The limited agreement between the two suPAR tests indicates that the RDT cannot replace or be used interchangeably with the ELISA, a conclusion that is likely to remain valid for other biomarkers and assays unless strong agreement has been demonstrated. Consequently, results from studies utilising different suPAR assays should not be pooled. If the tests are adopted for routine use, cut-offs associated with particular clinical management decisions (for example, admission or discharge from the emergency department) or weightings within multivariable triage tools (prediction models) should be assay-dependent [11,12,22]. Similarly, if suPAR measurements are used to inform participant recruitment into clinical trials, it is important that eligibility criteria are tailored to the assay used for enrolment [23].

The lack of agreement between the RDT and ELISA in our cohort is unexpected. The manufacturer reports that suPAR concentrations measured using the RDT should be within ±10% of measurements made on the ELISA, further underlining the need for context specific evaluation [13,22]. The impact of different detection methods for suPAR has been demonstrated

[24], however in our study both the ELISA and RDT used the same capture antibodies. Although RDT measurements were made after an additional freeze-thaw cycle, multiple studies have confirmed that suPAR concentrations are stable up to at least five repeated freeze-thaw cycles [25,26]. In our cohort, suPAR concentrations quantified using the RDT were higher than anticipated for non-severe patients attending an emergency department [13]. It is possible that an unknown factor may have interfered with the functioning of the RDT assay in this population and this merits further investigation. To our knowledge only one other study has quantified suPAR concentrations using the same RDT in India; 66.3% (126/190) of patients had suPAR concentrations > 5.5 ng/mL, however average concentration was not reported [27]. Nevertheless, as our results demonstrate, lack of agreement between inexpensive, quick, and practical RDTs and batched, quality-controlled, laboratory-based assays, does not necessarily preclude clinical utility of an index test on the field.

This is the first study evaluating the analytical performance and prognostic accuracy of a RDT for suPAR, head-to-head against a reference test in resource-constrained setting. The RDTs were performed on frozen plasma by laboratory technicians, rather than on fresh plasma as suggested by the manufacturer. If the tests were to inform real-time clinical decisions, fresh plasma would be used and trained laboratory technicians may not be available, especially in contexts with limited laboratory capacity. Future research should extend our results to explore the field-based implementation of the RDT using unfrozen patient samples and evaluate reliability and usability amongst lesser-trained practitioners.

We demonstrate comparable prognostic accuracy of a rapid, quantitative, point-of-care test for suPAR to the reference laboratory-based ELISA that was used to develop a previously validated clinical prediction model for the triage of patients presenting with moderate Covid-19. These results are promising and should encourage further exploration of the utility of suPAR-guided risk stratification of patients presenting with Covid-19 and other acute infections in settings with limited laboratory capacity. Lack of agreement between the two tests highlights the importance of undertaking evaluations of point-of-care tests to ensure cut-offs and weightings are adjusted accordingly prior to a test being recommended for clinical use.

## Supporting information

**S1 Text. Supplementary appendix.** Table A: STARD checklist. Fig A: Agreement between the suPAR RDT and ELISA. Fig B: Agreement between repeated RDT measurements. Text A: Prognostic accuracy of the suPAR RDT within the framework of a clinical prediction model. (DOCX)

## Acknowledgments

The authors thank Emmanuel Moreau at the Foundation for Innovative New Diagnostics (FIND) for his advice during the conduct of the study.

## Author Contributions

**Conceptualization:** Arjun Chandna, Sabine Dittrich, Jennifer Osborn, George M. Varghese, Yoel Lubell, Sakib Burza.

**Data curation:** Arjun Chandna, Raman Mahajan, Priyanka Gautam, Lazaro Mwandigha.

**Formal analysis:** Arjun Chandna, Lazaro Mwandigha.

**Funding acquisition:** Arjun Chandna, Sabine Dittrich, Jennifer Osborn, Yoel Lubell, Sakib Burza.

**Investigation:** Raman Mahajan, Priyanka Gautam, Vikash Kumar, Pragya Kumar, George M. Varghese.

**Methodology:** Arjun Chandna, Lazaro Mwandigha, Sabine Dittrich, Jennifer Osborn, Constantinos Koshiaris, George M. Varghese, Yoel Lubell, Sakib Burza.

**Project administration:** Arjun Chandna, Raman Mahajan, Sakib Burza.

**Supervision:** Pragya Kumar, Constantinos Koshiaris, George M. Varghese, Yoel Lubell, Sakib Burza.

**Visualization:** Arjun Chandna, Lazaro Mwandigha.

**Writing – original draft:** Arjun Chandna.

**Writing – review & editing:** Arjun Chandna, Raman Mahajan, Priyanka Gautam, Lazaro Mwandigha, Sabine Dittrich, Vikash Kumar, Jennifer Osborn, Pragya Kumar, Constantinos Koshiaris, George M. Varghese, Yoel Lubell, Sakib Burza.

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
