## [Editor Report · Decision Letter 0]

13 Dec 2022

  PGPH-D-22-01887 Point-of-care prognostication in moderate Covid-19: analytical validation and diagnostic accuracy of a soluble urokinase plasminogen activator receptor (suPAR) rapid test PLOS Global Public Health

Dear Dr. Chandna,

Thank you for submitting your manuscript to PLOS Global Public Health. As with all papers, your manuscript was reviewed by members of the editorial board. Based on our assessment, we have decided that the work does not meet our criteria for publication and will therefore be rejected.

Specifically, the Authors describe a point of care validation analysis for soluble urokinase plasminogen activator receptor (suPAR) compared to ELISA for diagnosing moderate COVID-19. The abstract lacks key information, and it is not clear what the aim or hypothesis was. Also, there was no control group included. No gold standard is mentioned

We are sorry that we cannot be more positive on this occasion. We very much appreciate your wish to present your work in one of PLOS's Open Access publications. Thank you for your support, and we hope that you will consider PLOS Global Public Health for other submissions in the future.

Yours sincerely,

Andrés F. Henao-Martínez, M.D.

Academic Editor
---

## [Decision Letter · Decision Letter 1]

20 Jul 2023

Point-of-care prognostication in moderate Covid-19: analytical validation and prognostic accuracy of a soluble urokinase plasminogen activator receptor (suPAR) rapid test

PGPH-D-22-01887R1

Dear Dr. Chandna,

We are pleased to inform you that your manuscript 'Point-of-care prognostication in moderate Covid-19: analytical validation and prognostic accuracy of a soluble urokinase plasminogen activator receptor (suPAR) rapid test' has been provisionally accepted for publication in PLOS Global Public Health.

Before your manuscript can be formally accepted you will need to complete some formatting changes, which you will receive in a follow up email. A member of our team will be in touch with a set of requests. Please kindly also address the minor comment made by reviewer #1 regarding the limitation of the study. 

Best regards,

Krutika Kuppalli, MD

Section Editor

Reviewer Comments (if any, and for reference):

Reviewer's Responses to Questions

**Comments to the Author**

1. If the authors have adequately addressed your comments raised in a previous round of review and you feel that this manuscript is now acceptable for publication, you may indicate that here to bypass the “Comments to the Author” section, enter your conflict of interest statement in the “Confidential to Editor” section, and submit your "Accept" recommendation.

Reviewer #1: All comments have been addressed

Reviewer #2: All comments have been addressed

2. Does this manuscript meet PLOS Global Public Health’s publication criteria? Is the manuscript technically sound, and do the data support the conclusions? The manuscript must describe methodologically and ethically rigorous research with conclusions that are appropriately drawn based on the data presented.

Reviewer #1: Yes

Reviewer #2: Yes

3. Has the statistical analysis been performed appropriately and rigorously?

Reviewer #1: Yes

Reviewer #2: Yes

4. Have the authors made all data underlying the findings in their manuscript fully available (please refer to the Data Availability Statement at the start of the manuscript PDF file)?

Reviewer #1: Yes

Reviewer #2: Yes

5. Is the manuscript presented in an intelligible fashion and written in standard English?

Reviewer #1: Yes

Reviewer #2: Yes

6. Review Comments to the Author

Reviewer #1: The manuscript is clear it is presentation and discussion of data. The findings are important and relevant. The revision has been done thoroughly. One minor suggestion could be to add a sentence that a limitation of the study is that the RDT measurements were conducted on frozen plasma and not on fresh plasma as suggested by the manufactor. (The reference to freeze/thaw stability was conducted on ELISA, not on RDT).

Reviewer #2: (No Response)

7. PLOS authors have the option to publish the peer review history of their article (what does this mean?). If published, this will include your full peer review and any attached files.

**Do you want your identity to be public for this peer review?** For information about this choice, including consent withdrawal, please see our Privacy Policy.

Reviewer #1: No

Reviewer #2: **Yes: **David Amoah Afrifah
